# 3,3'-Diindolylmethane induces apoptosis and autophagy in fission yeast

**Parvaneh Emami, Masaru Ueno** [ID] *

Graduate School of Integrated Sciences for Life, Hiroshima University, Higashi-Hiroshima, Hiroshima, Japan

* scmueno@hiroshima-u.ac.jp

## Abstract

3,3'-Diindolylmethane (DIM) is a compound derived from the digestion of indole-3-carbinol, found in the broccoli family. It induces apoptosis and autophagy in some types of human cancer. DIM extends lifespan in the fission yeast *Schizosaccharomyces pombe*. The mechanisms by which DIM induces apoptosis and autophagy in humans and expands lifespan in fission yeasts are not fully understood. Here, we show that DIM induces apoptosis and autophagy in log-phase cells, which is dose-dependent in fission yeast. A high concentration of DIM disrupted the nuclear envelope (NE) structure and induced chromosome condensation at an early time point. In contrast, a low concentration of DIM induced autophagy but did not disrupt NE structure. The mutant defective in autophagy was more sensitive to a low concentration of DIM, demonstrating that the autophagic pathway contributes to the survival of cells against DIM. Moreover, our results showed that the *lem2* mutant is more sensitive to DIM. NE in the *lem2* mutant was disrupted even at the low concentration of DIM. Our results demonstrate that the autophagic pathway and NE integrity are important to maintain viability in the presence of a low concentration of DIM. The mechanism of apoptosis and autophagy induction by DIM might be conserved in fission yeast and humans. Further studies will contribute to the understanding of the mechanism of apoptosis and autophagy by DIM in fission yeast and humans.

**Data Availability Statement:** All relevant data are within the manuscript and its Supporting Information files.

**Funding:** The authors received no specific funding for this work.

## Introduction

Apoptosis is a process that results in the death of damaged and unrepairable cells to maintain health in multicellular organisms [1]. It is a promising target for cancer therapy as it induces death in cancer cells [2]. Autophagy also recycles the cellular components; it can be induced by stress or nutrient starvation [3, 4]. Upregulation of autophagy extends the lifespan of the animal [5]. Autophagy induction can suppress the growth of cancer cells [6–9]; therefore, it has several potential benefits to human health, including longevity and cancer therapy.

3,3'-Diindolylmethane (DIM) is a compound derived from the digestion of indole-3-carbinol, found in the plants from the broccoli family, such as cabbage, broccoli, and rape [10]. Recent studies reported the anti-cancer effects of DIM through the induction of apoptotic cell death in breast cancer [11, 12], hepatoma [13, 14], prostate cancer [15, 16], or colon cancer [17–19]. DIM rapidly accumulates in the nuclear membranes (NM) of human breast

**Competing interests:** The authors have declared that no competing interests exist.

carcinoma (MCF-7) cells after 0.5–2 hours [20]. However, the outcome of the accumulation of DIM in NM remains unclear. DIM induces autophagy in gastric cancer cells [9] and in ovarian cancer cells [6] through endoplasmic reticulum (ER) stress. However, it is not completely understood how DIM induces ER stress.

Fission yeast *Schizosaccharomyces pombe* is a powerful model organism to study apoptosis [21–24] and autophagy [25, 26]. Apoptosis is detectable in fission yeast by observing dead cells and nuclear fragmentation [24, 27, 28] that could be induced by different conditions. Overexpression of endogenous gene such as *calnexin* in fission yeast causes apoptotic cell death, which partially depends on the Ire1, the ER-stress transducer [28]. On the contrary, *dga1 plh1* double mutant, which is defective in lipid metabolic pathway, promotes apoptosis in stationary-phase [29, 30]. Also, some compounds such as terpinolene [27] and alpha-thujone [21] induce apoptosis in this organism.

Autophagy can be induced in fission yeasts [31, 32] via different conditions. Nitrogen starvation induces autophagy in fission yeasts, which generates a nitrogen source for adaptation [33, 34]. Sulfur starvation also induces autophagy in fission yeasts [35]. Autophagy contributes to maintaining cellular viability in both nitrogen [25] and sulfur starvation [35]. Atg8 is critical for autophagosome formation in fission yeasts [8, 26] and used as a general autophagy marker that is distributed in the cytoplasm but forms bright foci when autophagy is induced by nitrogen starvation [31, 34, 36].

Nuclear envelope (NE) consists of the inner and outer NM, and the nuclear pore complex (NPC) in fission yeast [37]. One of the inner NM proteins in fission yeast is Lem2 that contributes to the NE integrity [38], maintaining the boundary between NE and ER [39] and controlling NM flow [40]. In *lem2Δ* cells, the distribution of ER protein such as Ish1 is affected [39]. Also, Lem2 recruits Cmp7 for NE closure by ESCRT-III (endosomal sorting complex required for transport-III) [41]. ESCRT-III is responsible for sealing a membrane rupture not only during mitosis but also during interphase to maintain NE integrity in fission yeast [39, 41, 42], which needs Vps4 to complete the sealing [41, 43].

There is not enough information about the effects of DIM on fission yeasts yet, except that it increases chronological lifespan (CLS) [44]. Here, we show that DIM decreases fission yeast viability at an early time point. We describe DIM as an apoptosis and autophagy inducer in this organism. Our results suggest that NE could be one of the early targets of DIM. We hope that these results help future studies, especially for cancer-related research.

## Materials and methods

### Procurement of yeast strains and construction of *htb1-GFP*, *GFP-atg8*, *ire1Δ*, *atg7Δ* and *lem2Δ* strains

Fission yeast strains used in our experiments are listed in Table 1.

Htb1-GFP strain was created by the introduction of NsiI partial-digested htb1GFPC plasmid, obtained from Dr. Matsuda and Dr. Hiraoka [45], to the FY7455 strain using *lys1*+ marker for selection.

To express GFP-tagged Atg8 in the wild-type strain (FY17186), we integrated EcoRI digested pHM43 plasmid containing a GFP-Atg8 gene with the *nmt41* promoter, gifted by Dr. Yamamoto [33], using *lys1*+ marker for selection.

To construct the *ire1Δ*::*Kanr* strain, we amplified the genomic DNA by polymerase chain reaction (PCR) using genomic DNA obtained from yPK002 as a template, a gift from Dr. Walter [46], and the primers 5′-TGGATGACTATACCCAAAGC-3′ and 5′-ATCCAACGATCC-CACAAGCG-3′. The resulting PCR product was introduced into the FY17186 strain by using

**Table 1. The strains used in this paper.**

| Name | Genotype | Source |
|---|---|---|
| **975** | $h^+$ | M. Yanagida |
| **FY7455** | $h^+$ leu1-32 ura4 -D18 his7 lys1-131 | NBRP |
| *htb1-GFP* | $h^+$ leu1-32 ura4-D18 his7 lys1$^+$::(hta1 htb1-GFP) | This paper |
| **FY17186** | $h^{90}$ ade6-216 leu1-32 lys1-131 ura4-D18 | NBRP |
| *GFP-atg8* | $h^{90}$ ade6-216 leu1-32 ura4- D18 lys1$^+$::Pnmt41-GFP-atg8$^+$ | This paper |
| *GFP-atg8 ire1Δ* | $h^{90}$ ade6-216 leu1-32 ura4- D18 lys1$^+$::Pnmt41-GFP-atg8$^+$ ire1::Kanr | This paper |
| *ire1Δ* | $h^{90}$ ade6-216 leu1-32 ura4- D18 lys1-131 ire1::Kanr | This paper |
| *atg7Δ* | $h^{90}$ ade6-216 leu1-32 ura4- D18 lys1-131 atg7-d1::Natr | This paper |
| **FY29360** | $h^{90}$ hat1$^+$::GFP-Kanr ark1::Kanr-Prad21-ark1$^+$ atg7-d1::Natr | NBRP |
| **yPK002** | $h^+$ ade6-M210 ura4-D18 leu1-32 ire1::Kanr | P. Walter |
| **YT2416** | $h^-$ lem2::Kanr | Y. Hiraoka |
| **80-G10** | $h^+$ ade6-210 leu1-32 cut11::cut11$^+$-GFP-HA-Kanr sid4$^+$-mCherry<<Natr | lab stock |
| **81-D02** | h+ leu1-32 cut11::cut11$^+$-GFP-HA-Kanr sid4$^+$-mCherry<<Natr lem2::Kanr | This paper |
| **1-1-1** | $h^+$ ade6-210 leu1 ura4 lys1 his7$^+$::lacI-GFP sod2.proximal[::Kanr-ura4$^+$-lacOp] sid4::sid4$^+$-GFP-natMX6 gar2::gar2$^+$-mCherry-hphMX6 | Ito *et al.* 2019 |
| **78-A02** | $h^+$ ade6-210 leu1 ura4 lys1 his7$^+$::lacI-GFP sod2.proximal[::Bsdr-ura4$^+$-lacOp] sid4::sid4$^+$-GFP-natMX6 gar2::gar2$^+$-mCherry-hphMX6 | This paper |
| **78-D01** | $h^+$ ade6-210 leu1 ura4 lys1 his7$^+$::lacI-GFP sod2.proximal[::Bsdr-ura4$^+$-lacOp] sid4::sid4$^+$-GFP-natMX6 gar2::gar2$^+$-mCherry-hphMX6 lem2::Kanr | This paper |

*S. pombe* Direct Transformation Kit Wako (FUJIFILM Wako, Osaka, Japan). The deletion was confirmed by a PCR.

A similar strategy was adopted to create the autophagy mutant *atg7Δ*, using the primers 5′–ATACGTAGAACTGCGGTGAG–3′ and 5′–CAAATGCAACTTCAGGATCC–3′ to amplify the mutated genomic DNA from *atg7-d1*::*Natr* strain (FY29360) and introduce it to the wild-type strain, FY17186. The deletion was confirmed by a PCR.

To construct *lem2Δ*::*Kanr* mutant for the viability assay, first, the strain named 78-A02 was created as a wild-type strain by replacing *Kanr* with *Bsdr* in the strain 1-1-1 shown in Ito et al. [47]. First, *Bsdr* fragment-containing plasmid was created by replacing the *Natr* gene in pNATZA13-mCherry-*atb2*$^+$, a gift from Dr. Y. Watanabe and Dr. T. Sakuno [48] with *Bsdr* gene by ligation of SacI and BglII digested pNATZA13-mCherry-*atb2*$^+$ with *Bsdr* gene, which was amplified from pSVEM-Bsdr as a template, a gift from Dr. A. Stewart [49], with the primers 5′–CCCCTCACAGACGCGTCACTCAACCCTATCTCGG–3′ and 5′–ATCCGCCGGTACGCGTCTCGAAATTAACCCTCAC–3′, using In-Fusion HD Cloning Kit (Takara bio, Shiga, Japan). In the next step, the DNA fragment containing the *Bsdr* gene was amplified using 5′–GACATGGAGGCCCAGAATAC–3′ and 5′–TGGATGGCGGCGTTAGTATC–3′. The resulting DNA fragment was used for transformation to replace *Kanr* with *Bsdr* in the strain 1-1-1. The resulting strain 78-A02 was used to delete with *lem2Δ*::*Kanr*. *lem2Δ*::*Kanr* DNA fragment was amplified using the primers: 5′–CCCTAATGATCATGGATTCTGT–3′ and 5′–ACTATGGATGCCTATTTTCCC–3′ using genomic DNA of the YT2416 strain, obtained from Dr. Hiraoka [50], as a template. Then, the PCR fragment was introduced to the strain 78-A02 to create strain 78-D01. To construct *cut11-GFP* strain with *lem2Δ*::*Kanr*, the lab stock strain (80-G10) was mated with the *lem2Δ*::*Kanr* strain YT2416, resulting in strain 81-D02.

## Viability assay by spot test

In viability assay, 3% glucose Yeast extract with adenine (YEA) liquid medium, and YEA plates were employed. A composition of 0.5% yeast extract, 3% glucose, and 40 mg/ml adenine was used for the liquid YEA medium or the YEA plate with 2% agar. For viability assay, the fission yeast cells were grown in an 8 ml liquid YEA medium overnight at 30°C (12～15 hours) to get the log-phase cells (0.5～$1 \times 10^7$ cells/ml) designated as day 0. As seen in Fig 1A, the culture was divided into two flasks. The same volume of Dimethyl sulfoxide (DMSO) or 20 μg/ml DIM, purchased from Combi-Blocks (San Diego, USA) dissolved in DMSO were added to each flask. On day 1, which was 24 hours after incubation with the drugs, cell number was adjusted to $1 \times 10^7$ cells/ml and used for viability assay with five-fold serial dilutions. The spotted plates were incubated at 30°C for 3 to 5 days to check cell viability. Every 24 hours, the viability assays were repeated.

To compare the effects of DIM on stationary-phase cells with that on log-phase cells, day0 cells, shown in Fig 1A, were cultured for 24 hours, which contained about ~$1 \times 108$ cells/ml designated as day 1. Then, DIM or DMSO were added (Fig 1C). As a control, DIM or DMSO were added to the day0 culture. In viability assay, as seen in Fig 1A, the cell number was adjusted to $1 \times 107$ cells/ml and used for viability assays with five-fold serial dilutions. Every 24 hours, the viability assay was repeated. The viability assays, shown in Figs 6A and 6B and 7A, were performed as shown in Fig 1A on fresh YEA plates containing 5 μg/ml DIM or 10 mM 2-mercaptoethanol (2-ME).

## Acute viability assay by spot test

The log-phase and stationary-phase cells were prepared as shown in Fig 1A and 1C. The cell concentrations were adjusted to $1 \times 10^7$ cells/ml. Immediately, DMSO or DIM (20 and 40 μg/ml) were added to the cultures and incubated at 30°C for 10 minutes. After washing the drugs, 100 μl of the same concentration ($1 \times 10^7$ cells/ml) of treated cultures were used for viability assay with five-time serial dilutions on YEA plates (Fig 2).

## Quantitation of the percentage of viability

The overnight culture for each strain was prepared by liquid YEA medium containing 3% of glucose and treated based on the desired experiment in three independent replications. After counting the cells number, the same cell number (500 cells) for each treatment were dispersed on YEA plates and three days later, the number of the grown colonies were counted and the viability percentages were calculated. The percentage of viability at day 0 (= reaching to the log-phase) was set to 100%, and subsequent grown colonies number was normalized to day 0 when the drugs were added to the log-phase cells (Figs 1B, 2B, 6C and 7B). For DIM effects on stationary-phase cells, the percentage of viability was normalized at day 1 (= reaching to the stationary-phase), and the viability percentage for day 1 was set to 100% and subsequent grown colonies number was normalized to day 1 (Figs 1D and 2C). Results for percent of viability for *atg7Δ*, *ire1Δ*, and *lem2Δ* strains compared to wild type have been measured three days after adding DMSO, DIM, or 2-ME to the log-phase liquid cultures.

## Apoptosis detection by acridine orange/ethidium bromide (AO/EB) and 4',6-diamidino-2-phenylindole (DAPI)

Dual staining by AO/EB was conducted to detect dead cells as previously described and terpinolene (300 mg/l) was used as control to show apoptosis phenotype [27]. Briefly, after precipitation and washing the cell with PBS (pH: 7.4), cells were resuspended in 100 μl PBS with 5 μl

## Viability assay in log-phase cells

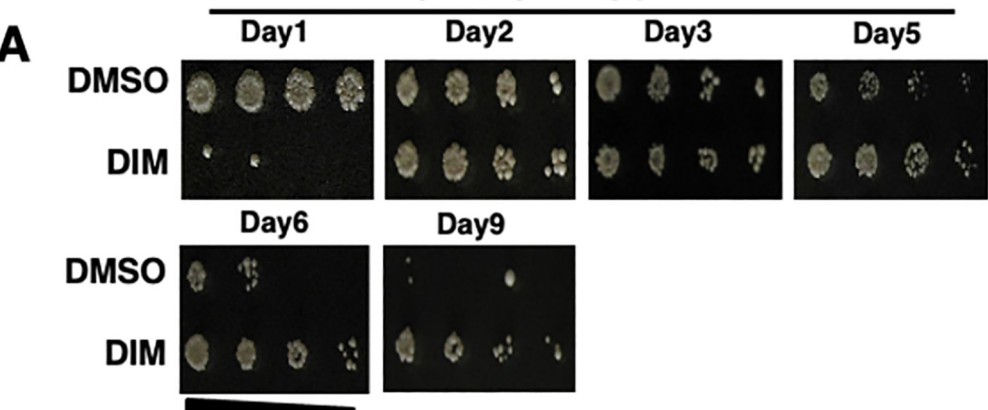

## Viability of log-phase cells after treatmet

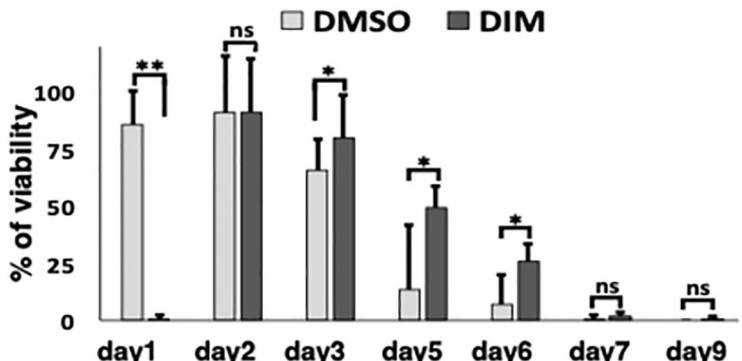

## Viability assay in stationary-phase cells

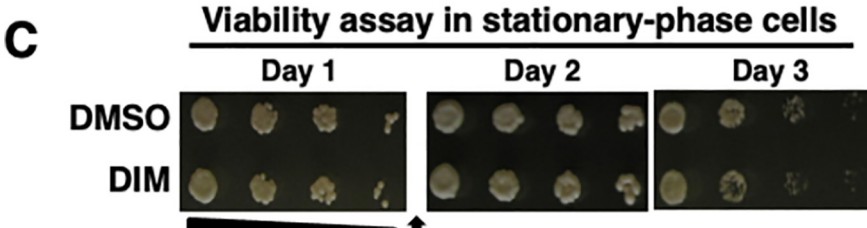

## Viability of stationary-phase cells after treatmet

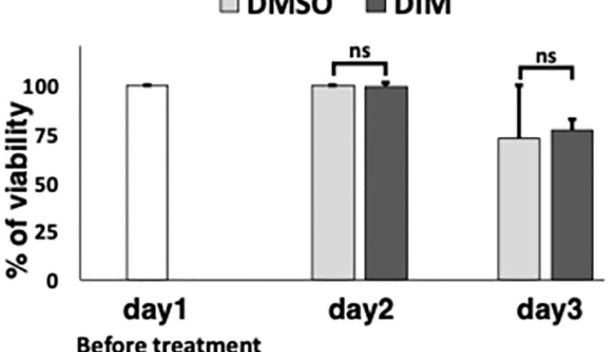

**Fig 1. DIM inhibits the cell viability of log-phase cells.** (A and B) 20 μg/ml 3,3'-Diindolylmethane (DIM) was added to the log-phase cells (~0.5–1 x 107 cells/ml) and the viability was studied by spot assay with five-fold serial dilutions (A). Dimethyl sulfoxide (DMSO) was used for the control experiment. The same cell concentration was spotted on YEA plates to regularly check the cell viability till day 9. Day 1 means 24 hours incubation after adding the drug. Percentages of viability for each day are shown (B). (C and D) 20 μg/ml DIM was added to the stationary-phase cells and the viability was studied by a spot assay (C). Log-phase cells were cultured for 24 hours. Then, DIM was added to the stationary-phase cells (⌒1 x 108 cells/ml). The same cell concentrations were spotted on YEA plates to regularly check the cell viability till day 3. Arrow shows the time of adding DIM. Percentages of viability of DIM treated stationary-phase cells are shown in D. See materials and methods for details.

of AO/EB mixture (AO 60 μg/ml: EB 100 μg/ml). Five minutes after incubation in the dark-room at room temperature, cells were washed twice with PBS and imaged by a Zeiss GFP filter set 38 HE of fluorescence microscope for AO, and a Zeiss mRFP filter set 63 HE for EB. Because of membrane integrity loss in dead cells, EB is up taken by dead cells only, but AO permeates into both dead and live cells and stains them green. Finally, dead cells are detectable in orange color in the merged images. To visualize nuclear fragmentation and condensation, cells were fixed by 70% ethanol for 20 minutes and washed with water. Then, precipitated cells were mixed with DAPI (0.1 mg/ml) in a 1:1 portion ratio to stain chromosomes. Stained cells were transferred on a glass slide and nuclei were observed under a fluorescence microscope. For microscopic analysis, a Zeiss microscope and the AxioVision 4.8 software were used to capture images, which were then analyzed using the ImageJ software.

### Nitrogen starvation and the condition in which DIM induces autophagy

The overnight culture in PMG medium (Edinburgh minimal medium (EMM) with 2% glucose and 3.75 g/L glutamate substituted for NH₄Cl as a nitrogen source with the supplementary nucleotides and amino acids with adjusted pH ~ 6) [51] was used to get log-phase cells of the GFP-Atg8 strain. Cells from 4 ml of the culture were precipitated and washed, and medium was replaced by EMM with 1% glucose without any source of the nitrogen and incubated at 30˚C. In parallel, 5 μg/ml DIM and DMSO were added to 4 ml of the overnight cultures in PMG (~0.5×10$^7$ cells/ml) and incubated at 30˚C in PMG. GFP-Atg8 foci were captured by a fluorescence microscope after four hours of nitrogen starvation and two hours of incubation with DIM, respectively.

### Statistical analysis

The post hoc T-test with Bonferrori correction was used to compare between the treatments or genotypes. All of the statistical analyses were done in excel software.

## Results

### DIM reduces cell viability in log-phase cells, but not in stationary-phase cells

A high concentration of DIM (20 μg/ml) increases lifespan while a low concentration of DIM (4 μg/ml) seems to decrease lifespan in fission yeasts [44]. In humans, DIM induces apoptosis and autophagy. These facts imply that DIM may have both positive and negative effects on cell viability in fission yeast. To study the effect of DIM on fission yeast in detail, a viability assay was performed by focusing on the log-phase stage. As shown previously [44], here also, the cell viability was better when cells were incubated for 9 days in the presence of 20 μg/ml DIM. Surprisingly, we found a severe growth defect when cells were cultured for only 24 hours in the presence of DIM (Fig 1A and 1B, S1 Fig). DIM did not inhibit growth after additional incubation, suggesting that either DIM does not decrease the viability when cells are in the

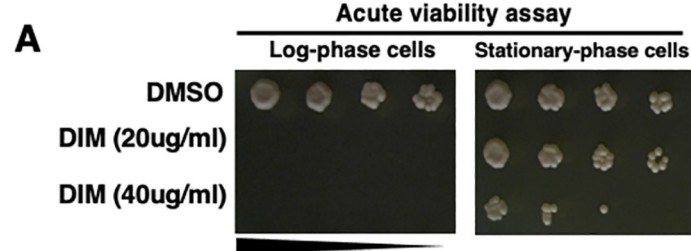

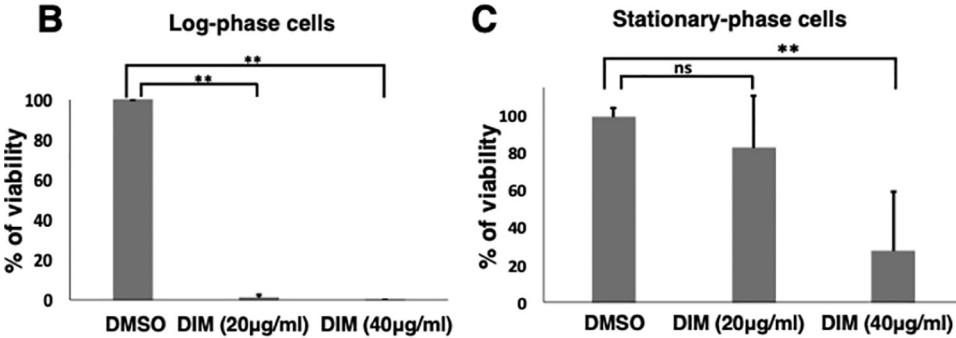

**Fig 2. 10 minutes incubation with DIM is enough to kill log-phase cells.** (A) To perform the acute viability assay, the same concentration of log-phase cells and stationary-phase cells (1 x 10⁷ cells/ml) were incubated in the presence of 3,3'-Diindolylmethane (DIM) or DMSO for 10 minutes. Treated cultures were spotted and incubated at 30°C for 3 to 5 days to capture images. (B and C) The percentages of viability for treated cells as in A are shown. See materials and methods for details.

stationary-phase or DIM is decomposed. To test the possibility that DIM does not decrease the viability when cells are in the stationary-phase, we added DIM to the cells that were in stationary-phase. We found that DIM did not decrease the viability when we added DIM to the cells in stationary-phase (Fig 1C and 1D).

## Acute effect of DIM to log-phase and stationary-phase cells

One of the explanations for the DIM resistance of the stationary-phase cells is that the cell number is high in the stationary-phase, which may result in a low effective concentration of the DIM relative to each cell. To test this possibility, we performed an acute viability assay (Fig 2). We adjusted cells to the same number between stationary- and log-phase cells and incubated them with DIM. To avoid cell phase-shifting after dilution of stationary-phase culture, we incubated cells for only 10 minutes with DIM. Our result showed that 10 minutes of incubation with DIM was enough to trigger cell death in log-phase cells (Fig 2A and 2B). In contrast, the stationary-phase cells with the same cell concentration were more resistant to DIM than the log-phase cells (Fig 2A and 2C). However, even the higher concentration (40 μg/ml) of DIM killed the stationary-phase cells (Fig 2A and 2C), demonstrating that DIM affects viability of stationary-phase cells when the DIM concentration is very high.

## DIM induces apoptosis in log-phase cells

Based on the recent results from human studies, DIM induces apoptosis in diverse cancer cells by different mechanisms [52, 53]. We assumed that cell killing by DIM in fission yeast may be due to apoptosis induction. To know if DIM induces apoptosis in fission yeast, first, we

confirmed that DIM induces cell death by staining dead and live cells (see materials and methods [27]). Dual staining by AO and EB showed that about 100 percentage of both log-phase and stationary-phase cells were killed by terpinolene, which is shown to induce apoptosis [27] (Fig 3A and 3B). We found that about 80 percent of the log-phase cells were killed by DIM after 20 hrs. In contrast, dead cell abundance in treated and control cells in stationary-phase cultures did not change (Fig 3A and 3B). These results show that DIM kills log-phase cells but not stationary-phase cells. Nuclear fragmentation is a hallmark of apoptotic cell death in fission yeasts [29]. To investigate whether DIM induces apoptosis in fission yeast, the nuclear shape was analyzed by DAPI staining. Apoptosis-inducing agent terpinolene induced nuclear fragmentation (Fig 3C and 3D) [27]. We detected nuclear fragmentation after six hours of incubation with DIM when cells are in log phase, suggesting that DIM induce apoptosis (Fig 3C and 3D). In contrast, we did not observe any difference in nuclei morphology in treated and control conditions in stationary-phase cells. Therefore, our results proved that DIM induces apoptosis in the log-phase cells but not in the stationary-phase cells.

## DIM induces nuclear condensation and NE disruption in log-phase cells within 10 minutes

To understand how DIM caused the reduction in viability, we analyzed nuclear morphology using cells expressing histone H2B fused to GFP (Htb1-GFP) in the presence of DIM. We observed that more than 98 percent of the log-phase cells showed nuclear condensation when the cells were incubated for 10 minutes in the presence of 20 μg/ml DIM (Fig 4A and 4D). We also analyzed NE morphology using a strain expressing Cut11-GFP in the same condition to visualize nuclear morphology. Clearly, DIM disrupted NE in log-phase cells within ten minutes of treatment (Fig 4B and 4E). These changes were not observed when we added DIM to the stationary-phase cells (Fig 4C–4E). Our results show that DIM induces nuclear condensation and NE disruption in log-phase cells, but not in stationary-phase cells within 10 minutes, which is much faster than nuclear fragmentation.

## Autophagy is induced by DIM, and the autophagic pathway but not the ER stress response pathway is required for the resistance to DIM

We showed that DIM leads to apoptosis in the log-phase cells of fission yeast (Fig 3). DIM induces autophagy in human cancer cells [6]. To understand if DIM also induces autophagy in fission yeast, first, we used a GFP-Atg8 expressing strain. It is reported that GFP-Atg8 foci are produced by nitrogen starvation in fission yeasts [34] (Fig 5A), which is a hallmark of autophagy induction. We observed GFP-Atg8 foci when log-phase cells were incubated with 5 μg/ml DIM for two hours (Fig 5B). This result suggested that DIM induces autophagy in fission yeast at a low concentration (5 μg/ml). Autophagy contributes to cell viability under nitrogen starvation [25]. We next investigated whether autophagy contributes to survival in the presence of DIM. As shown previously, *atg7Δ* cells that have a defect in the autophagic pathway lose viability under nitrogen starvation [54] (S2 Fig). We found that *atg7Δ* cells are more sensitive to low concentration of DIM (5 μg/ml) than wild-type cells (Fig 6A and 6C).

DIM also induces ER stress and accumulates Ire1, which is responsible for autophagy induction in human ovarian cancer cells [6]. The next question was whether ER stress-response contributes to the viability in the presence of 5 μg/ml DIM in fission yeast. As Ire1 is the major player for ER stress response (UPR regulation) in fission yeast [28, 55], we used *ire1Δ* cells to check the DIM sensitivity. We found that *ire1Δ* cells were not sensitive to DIM, while they were sensitive to the ER stress condition induced by 2-mercaptoethanol (Fig 6B and 6C). This data shows that ER stress response does not contribute to survival in the presence of

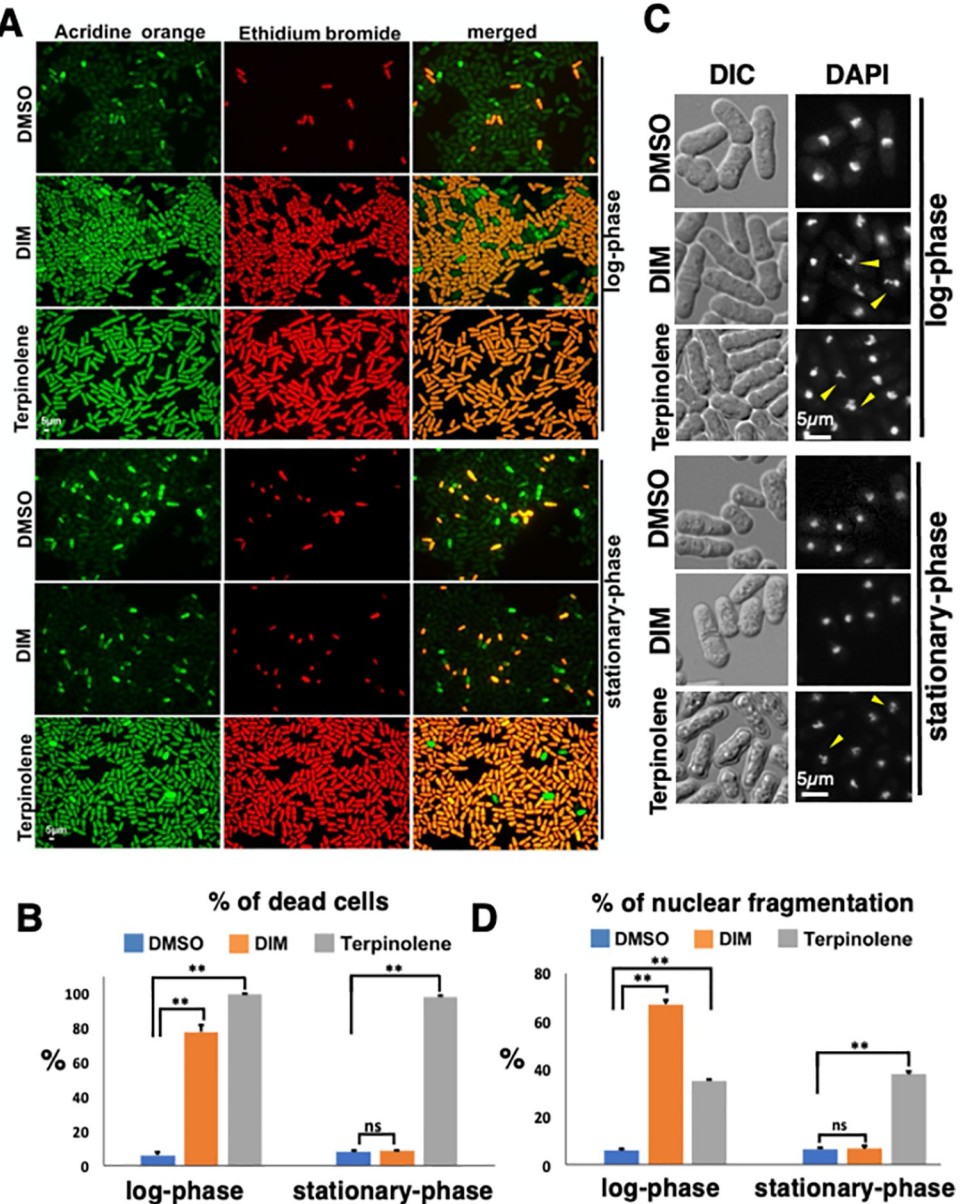

**Fig 3. DIM induces apoptosis in fission yeast log-phase cells.** (A and B) 20 hours after adding 20 μg/ml 3,3'-Diindolylmethane (DIM) to the log-phase and stationary-phase cells, the cells were stained with acridine orange (AO) and ethidium bromide (EB). Log-phase and stationary-phase cells were obtained as shown in Fig 1. 300 mg/l terpinolene was used as a control. See materials and methods for detailed condition of imaging (A). The percentages of dead cells were calculated from the merged images using ImageJ software were shown in B. At least 300 cells were counted with three independent experiments. (C and D) 300 mg/l terpinolene or 20 μg/ml DIM was added to the log-phase and stationary-phase cells. 4,6-diamidino-2-phenylindole (DAPI) staining was used to detect the nuclear fragmentation after six hours of incubation with DIM or terpinolene (C). The percentages of nuclear fragmentation are shown in D for at least 200 cells were with three independent experiments.

DIM. However, ER stress inhibitor, mithramycin, blocks DIM mediated ER stress and subsequently inhibits autophagy induction, suggesting that ER stress response is required for autophagy induction in ovarian cancer cells [6]. Therefore, we investigated if ER stress response is required for autophagy induction by DIM in fission yeast. GFP-Atg8 foci could still be

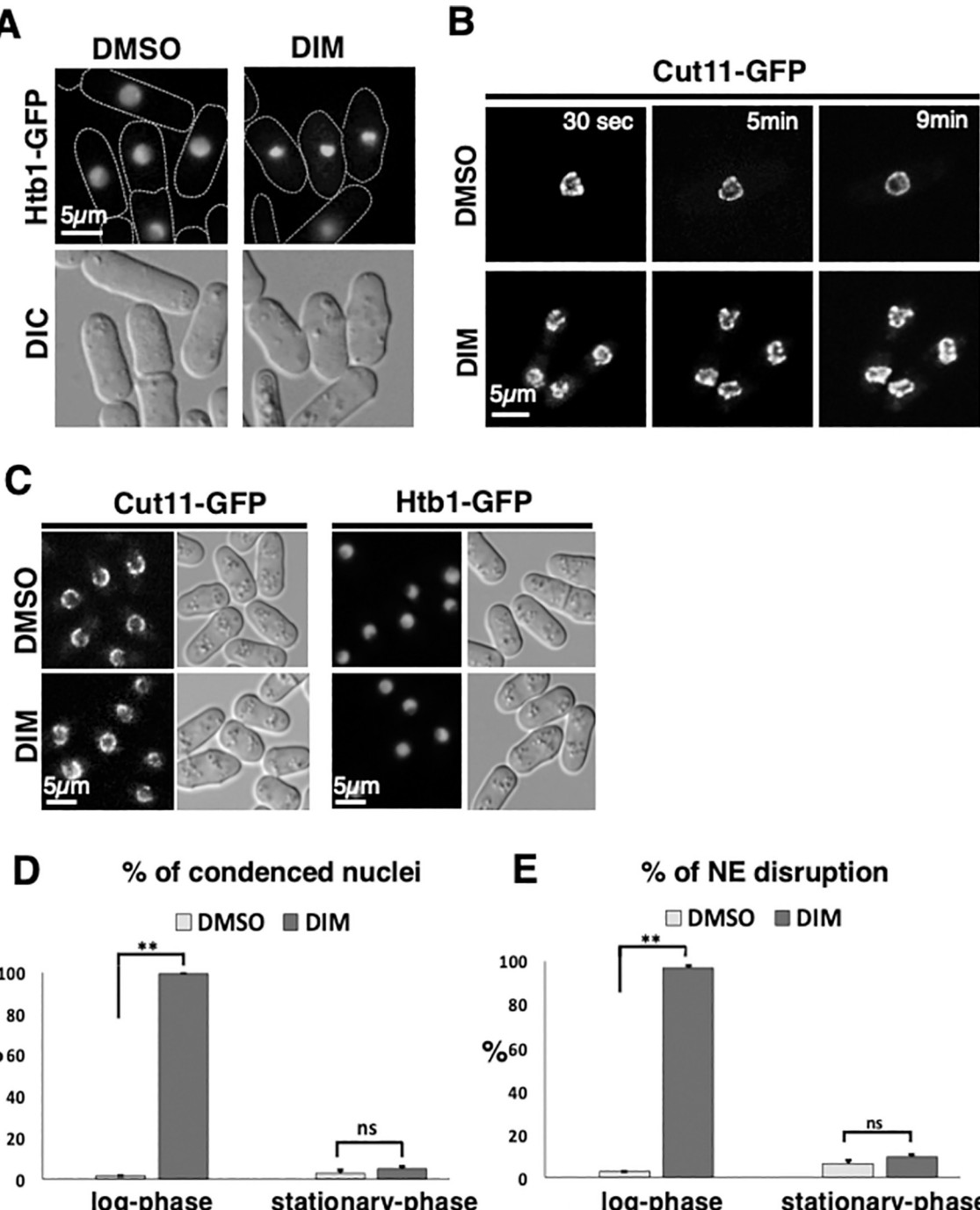

**Fig 4. DIM induces nuclear condensation and NE disruption in log-phase cells.** (A) 20 µg/ml 3,3'-Diindolylmethane (DIM) was added to log-phase cells (0.5〜1 x 10^7 cells/ml) expressing Htb1-GFP as a histone marker. Images taken after 10 min of incubation with DIM and DMSO are shown. (B) The time-lapse images of Cut11-GFP signal used as a marker for nuclear envelope (NE) are shown. Time after 20 µg/ml DIM addition to log-phase cells is shown. (C) 20 µg/ml DIM was added to the stationary-phase cells (10〜15 x 10^7 cells/ml) that are cultured as in Fig 1C. After 10 min, the shape of the nuclei and NE were observed as in A and B. (D and E) The percentages of condensed nuclei in D and disrupted NE in E are shown for treated cells in A-C. At least 200 cells were counted with three independent experiments for D and E separately.

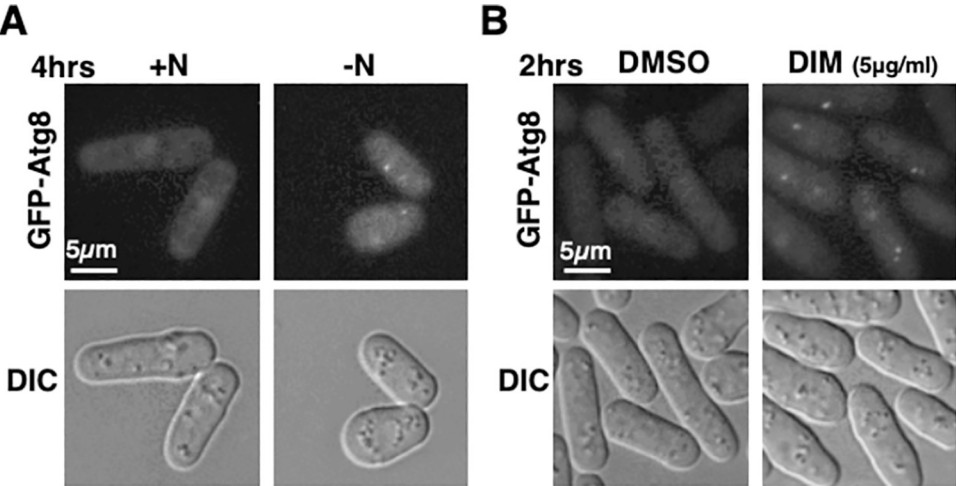

**Fig 5. A low concentration of DIM induces autophagy in fission yeasts.** (A and B) Cells expressing GFP-Atg8 were incubated for four hours in nitrogen starvation condition as a control (A) or two hours in the presence of a low concentration (5 µg/ml) of 3,3'-Diindolylmethane (B). GFP-Atg8 foci were used as markers for autophagy induction. See materials and methods for detailed condition of cell cultures.

observed in the *ire1Δ* strain in the presence of DIM (Fig 6D). These results suggested that, in contrast to the human case, ER stress response is not required for autophagy induction by a low concentration of DIM in fission yeast.

## Nuclear membrane protein, Lem2, is required for the resistance to DIM

We found that DIM disrupts NE rapidly (less than 10 minutes) (Fig 4B), implying that NE could be a direct target or NE disruption might be an early event in the response to DIM. In these cases, the mutant, which has a defect in NM might be more sensitive to DIM. Lem2 is an inner NM protein and plays an important role in regulating NE membrane homeostasis [56] and chromatin anchoring to the nuclear periphery in fission yeast [38]. Therefore, we investigated whether *lem2Δ* cells are more sensitive to DIM. We found that *lem2Δ* cells are more sensitive to a low concentration of DIM (Fig 7A and 7B). We also checked the NE morphology in the *lem2Δ* mutant in the presence of a low concentration of DIM. Unlike the high concentration of DIM (20 µg/ml), the low concentration of DIM (5 µg/ml) did not affect the NE of the wild-type strain (Fig 7C and 7D, S3 Fig). In contrast, NE in the *lem2Δ* mutant was significantly disrupted within four hours of incubation with DIM. These results confirmed that the NM protein, Lem2, is required for the resistance to DIM.

## Discussion

### DIM induces apoptosis in fission yeasts

Recent studies in humans show that DIM is a potential anti-cancer drug that acts by the induction of apoptosis in a wide range of cancer types including the breast [11, 12, 52, 57], prostate [16], gastric [58], pancreatic [59], and hepatoma [13]. DIM also induces autophagy that substantially suppresses tumor growth and causes cancer cell death [58, 60–62]. However, the mechanisms of autophagy and apoptosis induction by DIM in humans are not fully understood.

Here we demonstrated that DIM (20 µg/ml) kills most of the log-phase cells, but not the stationary-phase cells (Fig 3A and 3B). Additionally, we found that DIM induces nuclear

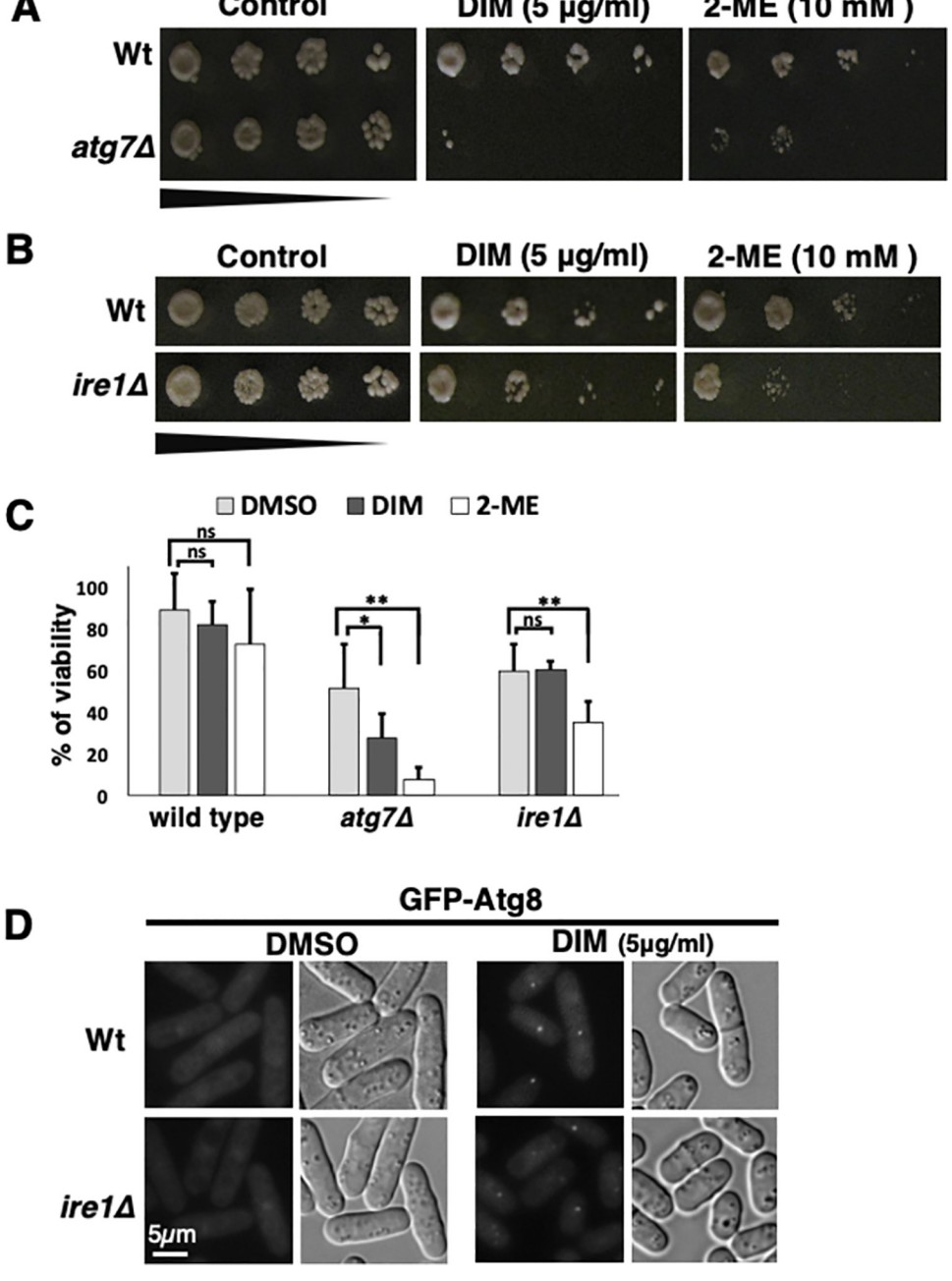

**Fig 6. Autophagy pathway, but not ER stress response pathway, is required for the resistance to DIM.** (A-C) Viability of log-phase cells of wild-type (Wt), *atg7Δ* (A) and *ire1Δ* (B) were studied by spot assay with five-fold serial dilutions in the presence of 5 µg/ml 3,3'-Diindolylmethane (DIM) or 10 mM 2-mercaptoethanol (2-ME). The images were taken 3–5 days after spotting. The percentages of viability are shown in C. (D) Autophagy induction by DIM was studied using *ire1Δ* mutant expressing GFP-Atg8. The experiment was performed as shown in Fig 5. See materials and methods for details.

fragmentation (Fig 3C and 3D). Our results for cell killing and nuclear fragmentation strongly suggested that DIM induces apoptosis at 20 µg/ml concentration (Fig 3). The acute viability assay showed that 10 minutes of treatment with 20 µg/ml of DIM is enough to trigger cell killing (Fig 2A and 2B). The cells were not stained with ethidium bromide just 10 minutes after

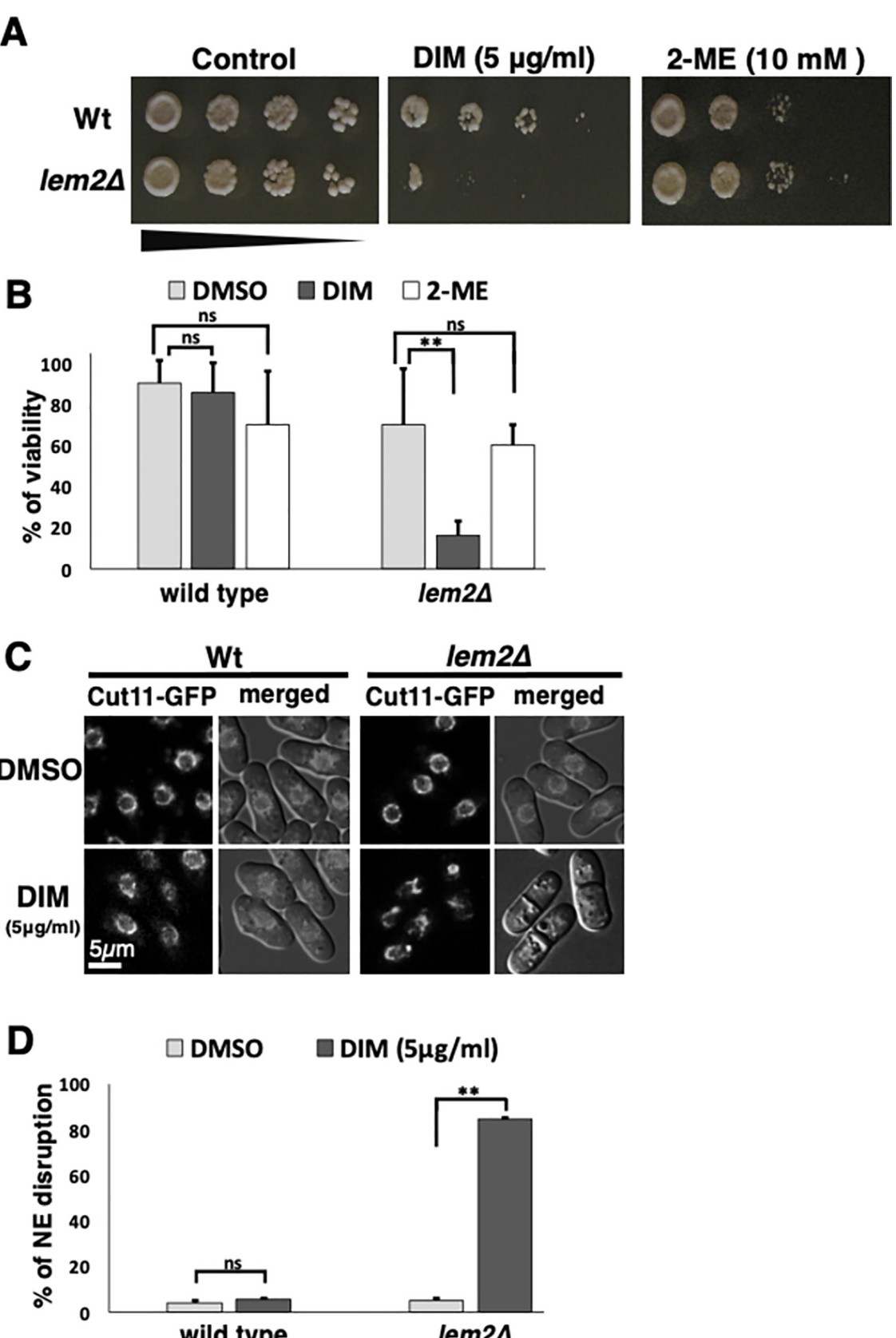

**Fig 7. Lem2 is required for the resistance to the low concentration of DIM.** (A and B) Viability of log-phase cells of wild-type and *lem2Δ* was studied by spot assay with five-fold serial dilutions in the presence of 5 μg/ml 3,3'-Diindolylmethane (DIM) or 10 mM 2-ME. (A) Spotting assay was done as described in Fig 6A. The percentages of viability are shown in B. (C and D) Fluorescent signals for Cut11-GFP protein in wild-type and *lem2Δ* log-phase cells were seen after 10 min or four hours incubation with 5 μg/ml DIM in YEA medium. Images merged with DIC are also shown in C. The percentages of NE disruption in wild type and *lem2Δ* are shown in D. At least 200 cells were counted with three independent experiments.

treating with 20 μg/ml DIM, suggesting that the cell wall remains intact and DIM does not kill the cells within 10 minutes (S4A Fig). However, this 10-minute treatment resulted in nuclear fragmentation after 6 hours (S4B Fig) and loss the viability (Fig 2A and 2B), suggesting that 10 minutes of incubation with DIM is enough to induce apoptosis and the process of apoptotic cell death continues even after DIM removal.

We found severe nuclear condensation (Fig 4A and 4D) and NE deformation (Fig 4B and 4E) when the log-phase cells were incubated with a high concentration of DIM for 10 minutes. To our knowledge, these phenotypes are not shown in fission yeast so far. We did not see nuclear fragmentation, nuclear condensation, or NE disruption, when we decreased the drug concentration to 5 μg/ml (S3 Fig). These results show that DIM effects are dose-dependent in log-phase cells. We speculate that DIM may inhibit some of the membrane proteins that are required for NE integrity, or inhibit control of the protein transport between nucleus and cytoplasm at the 20 μg/ml concentration. Therefore, by disrupting the NE integrity, DIM may cause the mis-localization of the nuclear or cytoplasmic protein. Subsequently, this mis-localization may change nuclear homeostasis and induce nuclear condensation. Nuclear condensation and NE disruption are the signs of apoptosis induction in fission yeast [28, 29, 63]. However, in our experiments, nuclear condensation and NE disruption happened much earlier than previous observations [28, 29, 63]. The disruption of nuclear transport and disassembly of the nuclear pore complex (NPC) occurs before caspase-9 activation in humans [64]. Therefore, it could be possible that the nuclear condensation and NE disruption by DIM in fission yeast, in the first 10 minutes of incubation, happen before apoptosis is induced. NE disruption by DIM and possible subsequent defect in protein transport between the nucleus and the cytoplasm might be a reason for apoptosis induction. However, it remains unclear how DIM induces apoptosis in fission yeast and further studies are necessary to understand the mechanism of apoptosis induction by DIM.

## Mechanism of autophagy induction by DIM

Similar to the results in human studies [6, 58], we found that DIM induces autophagy in fission yeast (Fig 5B) when we used the low concentration of DIM (5 μg/ml). Based on our data, the autophagy pathway contributes to the resistance to a low concentration of DIM (Fig 6A and 6C). Nitrogen starvation [32, 34], sulfur depletion [35], or ER stress [32] induce autophagy in fission yeast. It remains unclear how DIM induces autophagy in fission yeast. However, we found the autophagy induction by DIM is Ire1-independent, demonstrating that autophagy induced by DIM is not likely ER stress response-dependent. The *ire1Δ* strain did not show DIM sensitivity while it was ER stress-sensitive (Fig 6B and 6C). These results suggest that ER stress response is not required for the cell viability in the presence of a low concentration of DIM. DIM might mimic nitrogen starvation or sulfur depletion or alternatively, DIM might induce autophagy by an unidentified pathway.

## Lem2 is required for NE integrity in the low concentration of DIM

Finally, we showed that the inner NM protein, Lem2, is critical for cellular viability at a DIM concentration that induces autophagy (Fig 7A and 7B). The concentration of DIM that

induced autophagy resulted in NE disruption in *lem2Δ* cells but not in wild-type cells (Fig 7C and 7D). Lem2 acts as a barrier to membrane flow between the NE and other parts of the cellular membrane system [40]. On the other hand, the amount of C24:0 fatty acid, which is important for the survival of yeast cells [65], is reduced in the absence of both Lem2 and Bqt4 in fission yeast [66]. Therefore, the physical property of the NM including fatty acid composition and possibly membrane protein composition may be affected in *lem2Δ* cells. This could be one of the possible reasons why NE in *lem2Δ* cells is more susceptible to the lower concentration of DIM. Also, *lem2Δ* cells have defects in the recruitment of proteins such as Vps4 and Cmp7 to NE and in repairing the holes of NE by ESCRT-III [39]. Therefore, DIM may make holes in the NE of wild-type cells that could be repaired. In contrast, in the *lem2Δ* strain with defects in Vps4 expression and Cmp7 localization, ESCRT-III may be unable to seal the NE. This could be another reason for the DIM sensitivity in *lem2Δ* cells.

## Stationary-phase cells are resistant to DIM

Our result showed that there is a difference between log-phase and stationary-phase cells in response to DIM (Figs 1 and 2). We found that the log-phase cells, but not stationary-phase cells, are sensitive to the higher concentration of DIM (20 μg/ml). The ratio of drug molecules per cell is not the reason for the difference in viability of log-phase and stationary-phase cells in response to DIM (Fig 2). Also, NE, nuclear morphology, and cell survival were not affected at the higher concentration of DIM (20 μg/ml) in stationary-phase cells (Figs 3 and 4).

Our results suggest that NE could be one of the early targets for DIM in log-phase-cells (Fig 4B). Therefore, the difference in the nature of NE and protein composition in the NE between log-phase and stationary-phase cells could be the reason for the difference in sensitivity. The NE protein Ish1 has a higher level of expression in stationary-phase than in log-phase [67]. Therefore, Ish1 may be a candidate for the resistance to DIM in stationary-phase cells. However, it is possible that DIM targets the unknown NE-independent protein(s) in log-phase cells, which may be absent in stationary-phase cells.

In conclusion, we showed for the first time that DIM induces apoptosis and autophagy in fission yeast, which was previously reported in humans. The mechanism by which DIM induces apoptosis and autophagy may be conserved in yeast and humans, which helps to study it easily in a unicellular organism such as the fission yeast. Also, the difference between apoptosis and autophagy induction due to DIM concentration implies the importance of the dose-dependent manner in which DIM affects the final killing or survival of the cells. Also, we showed that NE could be one of the early targets of DIM. The understanding of apoptosis and autophagy mechanism by DIM in fission yeast may be helpful for human cancer and longevity research. We think that study of the NE structure could be a good starting point for these cases.

## Supporting information

**S1 Fig. DIM inhibits cell growth in log-phase cells during the first 24 hours.** 20 μg/ml 3,3'-Diindolylmethane (DIM) was added to the log-phase cells and the cell growth was measured during the first day.
(TIF)

**S2 Fig. *atg7Δ* strain lose viability under nitrogen starvation.** Percentages of viability under nitrogen starvation for three days are shown. The log-phase cells of auxotrophic wild type strain ($h^{90}$ *ade6-216 leu1-32, ura4-D18, lys1-131*) and its *atg7Δ* mutant were cultured in YEA medium. Then cells were cultured in EMM without any nitrogen source and lysine, uracil,

adenine, and leucine. Day1 means that cells are cultured under nitrogen starvation for one day.
(TIF)

**S3 Fig. DIM does not induce nuclear condensation and fragmentation at a low DIM concentration.** (A) 5 μg/ml 3,3'-Diindolylmethane (DIM) was added to the log-phase cells expressing Cut11-GFP and Htb1-GFP and assayed after 10 minutes (B) DAPI staining was performed for log-phase cells after 5 hours of incubation with 5 μg/ml DIM.
(TIF)

**S4 Fig. DIM triggers cell death but cell membrane integrity is not destroyed at an early time point.** (A) The wild type log-phase cells were treated with 20 μg/ml DIM for 10 minutes. Dead cell abundance was detected by acridine orange (AO)/ ethidium bromide (EB) method as shown in orange color. The percentages of nuclear fragmentation are shown in B. The log phase cells are treated with 20 μg/ml or 40 μg/ml 3,3'-Diindolylmethane (DIM) for 10 minutes. After treatment, cells were washed, precipitated, and transferred to the sample tubes containing sold YEA medium. Six hours later, nuclear fragmentation was monitored by DAPI staining. At least 200 cells were counted with three independent experiments.
(TIF)

## Acknowledgments

We are grateful to Dr. Mitsuhiro Yanagida, Dr. Atsushi Matsuda, Dr. Yasushi Hiraoka, Dr. Peter Walter, Dr. Ayumu Yamamoto, Dr. Yoshinori Watanabe, and Dr. Takeshi Sakuno, Dr. A. Francis Stewart, and Sayaka Suzuki for the strains and plasmids, respectively. We are also thankful to Dr. Hizlan Hincal Agus for his helpful guidance.

## Author Contributions

**Conceptualization:** Masaru Ueno.

**Data curation:** Parvaneh Emami.

**Formal analysis:** Parvaneh Emami.

**Investigation:** Parvaneh Emami, Masaru Ueno.

**Methodology:** Parvaneh Emami, Masaru Ueno.

**Project administration:** Masaru Ueno.

**Resources:** Parvaneh Emami.

**Supervision:** Masaru Ueno.

**Visualization:** Parvaneh Emami.

**Writing – original draft:** Parvaneh Emami.

**Writing – review & editing:** Masaru Ueno.

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
