## [Decision Letter · Decision Letter 0]

24 Aug 2021

PONE-D-21-23534

3,3'-Diindolylmethane induces apoptosis and autophagy in fission yeast

PLOS ONE

Dear Dr. Ueno,

Thank you for submitting your manuscript to PLOS ONE. After careful consideration, we feel that it has merit but does not fully meet PLOS ONE’s publication criteria as it currently stands. Therefore, we invite you to submit a revised version of the manuscript that addresses the points raised during the review process.

Both reviewers felt that the area of investigation was of interest and recognize that there is a substantial body of work represented in the manuscript.  However, they had some concerns about the depth or rigorousness of the analysis of the study in its current form.  These include the quality of the dilution tests, validity of the drug effects for long-time incubation, as well as the lack of appropriate control strains in each experiment, statistical analysis, or the logic of the experiments.   If you decide to revise the work, please submit a list of changes or a rebuttal against each point that is being raised when you submit the revised manuscript. I believe that you can easily address these concerns by experiments or editing the texts.

We look forward to receiving your revised manuscript.

Kind regards,

Reiko Sugiura, M.D., PhD.

Academic Editor

PLOS ONE

Journal Requirements:

"No"

"NO authors have competing interests"

Reviewers' comments:

Reviewer's Responses to Questions

**Comments to the Author**

1. Is the manuscript technically sound, and do the data support the conclusions?

Reviewer #1: Yes

Reviewer #2: Partly

2. Has the statistical analysis been performed appropriately and rigorously? 

Reviewer #1: Yes

Reviewer #2: No

3. Have the authors made all data underlying the findings in their manuscript fully available?

Reviewer #1: Yes

Reviewer #2: Yes

4. Is the manuscript presented in an intelligible fashion and written in standard English?

Reviewer #1: Yes

Reviewer #2: Yes

5. Review Comments to the Author

Reviewer #1: In this manuscript, the authors examine the effect of 3,3'-diindolylmethane, which has potential anti-cancer properties, in the fission yeast S. pombe. High concentrations of DIM induce apoptosis-like cell death in log-phase cells, while stationary-phase cells are resistant to high dosage of DIM. DIM impairs nuclear envelope integrity, and a knock-out mutant of nuclear membrane protein Lem2 shows higher sensitivity to DIM than wild-type cells. DIM also induces autophagy at a low concentration, as shown in human cells.

Overall, the experiments are well conducted and the data presented is informative.

Specific points:

1. Figure 1A shows that DIM inhibits fission yeast growth only for 1 day. The authors speculate from this data that "DIM does not inhibit growth when cells are in the stationary-phase" (Lines 203-204). It is difficult to draw the conclusion from Figure 1A, while this speculation seems correct from the other data such as Figure 1B and C. Do surviving stationary-phase cells grow in the presence of DIM? It is likely that DIM toxicity is lost after day 2 in the experiment of Figure 1A.

The growth curve for the first 24 hours in the high concentration of DIM may be informative.

2. Figure 2C. Frequencies of cells with fragmented nucleus should be shown.

3. Lines 308-309 read that 10-minute DIM treatment can induce apoptosis. Does 10-minute incubation induce nuclear fragmentation?

Reviewer #2: SUMMARY: DIM induces apoptosis and autophagy in a fission yeast. The authors observe that a high concentration of DIM disrupted the nuclear envelope and induced chromosomal condensation. This did not happen with a low concentration of DIM. Instead the cells were triggered to undergo autophagy. A atg7 mutant defective in autophagy was more sensitive to low doses of DIM. Similarly, a lem2 mutant defective in NE physiology was also more sensitive to the drug.

Comments on the Figures:

Figure 1A: Providing two dilution spots is not typical. The authors should provide at least four dilution spots so that they can quantify the loss of viability. Provide statistical analysis. When the authors comment that DIM did not inhibit growth after additional incubation, suggesting that DIM does not inhibit growth when cells are in stationary phase, how do they rule out the alternative explanation that DIM simply degrades within 24 hours and such is not present to mitigate cell growth after 1 day. Do they know the half-life of DIM in a DMSO solution? How do the authors know that the DIM is still present at day 5 or day 9? 1.

Figure 1B: Once again, the authors need to provide at least four dilution spots so that they can quantify the viability of the cells and graph it. Provide statistical analysis. Also, this panel is not easy to understand. More labels are needed.

Figure 1C: Once again, the authors need to provide at least four dilution spots so that they can quantify the viability of the cells and graph it. Provide statistical analysis.

Figure 2: Can the authors include a positive control that is known to induce apoptosis in fission cells? Otherwise it is hard to know how to interpret the figures.

Figure 3: Can the authors include a positive control that is known to induce nuclear condensation and NE disruption in fission cells? Otherwise it is hard to know how to interpret the figures. Can the authors quantify the NE disruption in Figures 3C and 3D in the same way they quantified 3A in 3B? Provide statistical analysis.

Figure 5A: Can the authors include a positive control that is known to induce autophagy in fission cells to show lethality of the atg7 mutant? Also, can they quantify the loss of viability as they did in Figure 2B, for example. Provide statistical analysis.

Figure 5B: Why did the authors expect autophagy induction by DIM to be dependent on Ire1p? The authors need to describe their expectations that link ER-stress and autophagy in the manuscript.

Figure 6A: Can the authors quantify the loss of viability and graph it? Provide statistical analysis.

Figure 6B: Can the authors quantify loss of nuclear envelope integrity and graph it? Provide statistical analysis.

6. PLOS authors have the option to publish the peer review history of their article (what does this mean?). If published, this will include your full peer review and any attached files.

Reviewer #1: No

Reviewer #2: No

---

## [Author Response · Author response to Decision Letter 0]

15 Nov 2021

Response to Reviewer1 

Reviewer #1: In this manuscript, the authors examine the effect of 3,3'-diindolylmethane, which has potential anti-cancer properties, in the fission yeast S. pombe. High concentrations of DIM induce apoptosis-like cell death in log-phase cells, while stationary-phase cells are resistant to high dosage of DIM. DIM impairs nuclear envelope integrity, and a knock-out mutant of nuclear membrane protein Lem2 shows higher sensitivity to DIM than wild-type cells. DIM also induces autophagy at a low concentration, as shown in human cells.

Overall, the experiments are well conducted and the data presented is informative.

(1) Response to [1. Figure 1A shows that DIM inhibits fission yeast growth only for 1 day. The authors speculate from this data that "DIM does not inhibit growth when cells are in the stationary-phase" (Lines 203-204). It is difficult to draw the conclusion from Figure 1A, while this speculation seems correct from the other data such as Figure 1B and C. Do surviving stationary-phase cells grow in the presence of DIM? It is likely that DIM toxicity is lost after day 2 in the experiment of Figure 1A. The growth curve for the first 24 hours in the high concentration of DIM may be informative.]

(1-1) Response to: ["DIM does not inhibit growth when cells are in the stationary-phase" (Lines 203-204). It is difficult to draw the conclusion from Figure 1A.]

I agree to the reviewer’s comment. Therefore, we revised sentence as shown below.

Previous: DIM did not inhibit growth after additional incubation, suggesting that DIM does not inhibit growth when cells are in the stationary-phase.

Page 15 lines 226-231

Revised: DIM did not inhibit growth after additional incubation, suggesting that either DIM does not decrease the viability when cells are in the stationary-phase or DIM is decomposed. To test the possibility that DIM does not decrease the viability when cells are in the stationary-phase, we added DIM to the cells that were in stationary-phase.

(1-2) Response to: [Do surviving stationary-phase cells grow in the presence of DIM? It is likely that DIM toxicity is lost after day 2 in the experiment of Figure 1A.]

As suggested by reviewer, we asked whether the surviving stationary-phase cells grow in the presence of DIM. We found that the surviving stationary-phase cells cannot grow in the presence of DIM. This result suggests that either DIM is decomposed in 2 days or surviving stationary-phase cells tried to return to log-phase cell on DIM plate and became sensitive to DIM. Stephan et. al. Aging Cell (2013) shows that pka1 mutant is sensitive to DIM even after 3 days incubation with DIM, suggesting that DIM is still active after 3 days incubation. However, it is still possible that DIM could be partially decomposed after 2 days. Our main question is acute effect of DIM. Therefore, we will focus on acute effect of DIM. 

We have a hypothesis that log phase cells become resistant to DIM when cells are incubated in the presence of DIM after 24 h possibly by epigenetic adaptation. These DIM resistant cells can reach to stationary phase. This epigenetic adaptation is erased when the surviving stationary-phase cells are incubated in fresh YEA plate with DIM. This hypothesis will be studied in another manuscript. Therefore, we do not want to include data showing that the surviving stationary-phase cells can not grow in the presence of DIM in our current manuscript. 

(1-3) Response to: [The growth curve for the first 24 hours in the high concentration of DIM may be informative.]

As suggested reviewer, we added the growth curve for the first 24 hours in the high concentration of DIM in SFig. S1 and text in page 15 line 226). Cell number did not increase first 24 hours in the high concentration of DIM. 

(2) Response to: [2. Figure 2C. Frequencies of cells with fragmented nucleus should be shown.]

As suggested by reviewer, we added the frequencies of cells with fragmented nucleus in new Fig. 3D.

(3) Response to: [3. Lines 308-309 read that 10-minute DIM treatment can induce apoptosis. Does 10-minute incubation induce nuclear fragmentation?]

As suggested by reviewer, we testes this point and found that at 10 min incubation with DIM induce nuclear fragmentation after 6h. We added the frequencies of cells with fragmented nucleus in the new Fig. S4B to mention that 10 min incubation with DIM induce nuclear fragmentation after 6h. To explain this, we added sentence below.

Added sentence page 26 lines 425-429 

However, this 10-minute treatment resulted in nuclear fragmentation after 6 hours (S4B Fig) and loss the viability (Figs. 2A and B), suggesting that 10 minutes of incubation with DIM is enough to induce apoptosis and the process of apoptotic cell death continues even after DIM removal.

(4) Additional modification that was not suggested by reviewer #1.

We removed previous Fig. 1B log phase data, because it was a repetition of previous Fig. 1A. 

We also removed previous Fig. 6B 10 min data, because there was no change at all. Moreover, these data were the repetition of previous Fig. 1C and D. 

We also removed previous Fig. S1B data, because it was a repetition of previous Fig. 1C. 

Response to Reviewer2

Reviewer #2: SUMMARY: DIM induces apoptosis and autophagy in a fission yeast. The authors observe that a high concentration of DIM disrupted the nuclear envelope and induced chromosomal condensation. This did not happen with a low concentration of DIM. Instead the cells were triggered to undergo autophagy. A atg7 mutant defective in autophagy was more sensitive to low doses of DIM. Similarly, a lem2 mutant defective in NE physiology was also more sensitive to the drug.

Comments on the Figures:

(1) Response to: [Figure 1A: Providing two dilution spots is not typical. The authors should provide at least four dilution spots so that they can quantify the loss of viability. Provide statistical analysis. When the authors comment that DIM did not inhibit growth after additional incubation, suggesting that DIM does not inhibit growth when cells are in stationary phase, how do they rule out the alternative explanation that DIM simply degrades within 24 hours and such is not present to mitigate cell growth after 1 day. Do they know the half-life of DIM in a DMSO solution? How do the authors know that the DIM is still present at day 5 or day 9?]

(1-1) Response to [The authors should provide at least four dilution spots so that they can quantify the loss of viability.]

As suggested by reviewer, we provided four dilution spots in all data (Fig. 1A and C, Fig. 2A, Fig. 6A and B, Fig. 7A). 

(1-2) Response to [Provide statistical analysis]

As suggested by reviewer, we quantitated dada and provided statistical analysis for all data (Fig. 1B and D, Fig. 2B and C, Fig. 3B and D, Fig. 4D and E, Fig. 6C, Fig. 7B and D). 

(1-3) Response to: [When the authors comment that DIM did not inhibit growth after additional incubation, suggesting that DIM does not inhibit growth when cells are in stationary phase, how do they rule out the alternative explanation that DIM simply degrades within 24 hours and such is not present to mitigate cell growth after 1 day. Do they know the half-life of DIM in a DMSO solution? How do the authors know that the DIM is still present at day 5 or day 9?]

I agree to reviewer’s comment. Stephan et. al. Aging Cell (2013) shows that pka1 mutant is sensitive to DIM even after 3 days incubation with DIM, suggesting that DIM is still active after 3 days incubation. However, it is still possible that DIM could be partially decomposed after two days. Our main question is acute effect of DIM. Therefore, we will focus on acute effect of DIM. We revised sentence as shown below. 

Previous: DIM did not inhibit growth after additional incubation, suggesting that DIM does not inhibit growth when cells are in the stationary-phase.

Page 15 lines 226-231

Revised: DIM did not inhibit growth after additional incubation, suggesting that either DIM does not decrease the viability when cells are in the stationary-phase or DIM is decomposed. To test the possibility that DIM does not decrease the viability when cells are in the stationary-phase, we added DIM to the cells that were in stationary-phase.

(2-1) Response to: [Figure 1B: Once again, the authors need to provide at least four dilution spots so that they can quantify the viability of the cells and graph it. Provide statistical analysis. Also, this panel is not easy to understand.]

Same response as shown above in (1-1) and (1-2).

(2-2) Response to: [More labels are needed.]

As suggested by reviewer, we provided labels to show the difference between log phase and stationary phase cells in Fig. 1 and 2. To make Figure 1 more easily understandable, we separated data to new Fig.1 (chronic assay) and new Fig. 2 (acute assay). Moreover, log-phase data in previous Fig. 1B had repetition to previous Fig. 1A. Therefore, we removed log-phase data that was repeatedly shown in previous Fig. 1B. We also changed the position of spot assay picture from up-down to side-by-side with new labels (log-phase cells and stationary phase cells) in new Fig. 2A. 

(3) Response to: [Figure 1C: Once again, the authors need to provide at least four dilution spots so that they can quantify the viability of the cells and graph it. Provide statistical analysis.]

Same response as shown above in (1-1) and (1-2).

(4) Response to: [Figure 2: Can the authors include a positive control that is known to induce apoptosis in fission cells? Otherwise it is hard to know how to interpret the figures.]

As suggested by reviewer, we included a positive control called Terpinolene that is known to induce apoptosis and shown in new Fig. 3 and text page 18 line 278 and 285). 

(5)　Response to:　[Figure 3: Can the authors include a positive control that is known to induce nuclear condensation and NE disruption in fission cells? Otherwise it is hard to know how to interpret the figures.]

It is impossible to provide positive control that induce nuclear condensation and NE disruption within 10 minutes because there is no paper showing that nuclear condensation and NE disruption are induced 10 min. after drug addition. To our knowledge, this is completely new experiment and new data. Therefore, we added sentences shown below.

Added sentence to page 26 lines 432-433

To our knowledge, these phenotypes are not shown in fission yeast so far.

(6) Response to: [Can the authors quantify the NE disruption in Figures 3C and 3D in the same way they quantified 3A in 3B? Provide statistical analysis.]

Same response as shown above in (1-2).

(7) Response to: [Figure 5A: Can the authors include a positive control that is known to induce autophagy in fission cells to show lethality of the atg7 mutant?] 

Yes, we added data showing that atg7 mutant is more sensitive to nitrogen starvation in SFig. 2 and text page 21 line 345, because it is already published (Fukuda et al. 2020), we showed this data in SFig. 2. 

(8) Response to: [Also, can they quantify the loss of viability as they did in Figure 2B, for example. Provide statistical analysis.]

Same response as shown above in (1-2).

(9) Response to: [Figure 5B: Why did the authors expect autophagy induction by DIM to be dependent on Ire1p? The authors need to describe their expectations that link ER-stress and autophagy in the manuscript.]

It is reported that ER stress response is required for autophagy induction in ovarian cancer cells. Therefore, we investigated if ER stress response is required for autophagy induction by DIM in fission yeast. To explain it, we added following sentence.

Added sentence in page 23 lines 374-378

However, ER stress inhibitor, mithramycin, blocks DIM mediated ER stress and subsequently inhibits autophagy induction, suggesting that ER stress response is required for autophagy induction in ovarian cancer cells [6]. Therefore, we investigated if ER stress response is required for autophagy induction by DIM in fission yeast.

(10) Response to: [Figure 6A: Can the authors quantify the loss of viability and graph it? Provide statistical analysis.]

Same response as shown above in (1-2).

(11) Response to: [Figure 6B: Can the authors quantify loss of nuclear envelope integrity and graph it? Provide statistical analysis.]

Same response as shown above in and (1-2).

(12) Additional modification that was not suggested by reviewer 2.

We removed previous Fig. 6B 10 min data, because there is no change at all. Moreover, these data were the repetition of previous Fig. 1C and D. 

We also removed previous Fig. S1B data, because it was a repetition of previous Fig. 1C.

---

## [Decision Letter · Decision Letter 1]

25 Nov 2021

3,3'-Diindolylmethane induces apoptosis and autophagy in fission yeast

PONE-D-21-23534R1

Dear Dr. Ueno,

We’re pleased to inform you that your manuscript has been judged scientifically suitable for publication and will be formally accepted for publication once it meets all outstanding technical requirements.

Kind regards,

Reiko Sugiura, M.D., PhD.

Academic Editor

PLOS ONE

Additional Editor Comments (optional):

Reviewers' comments:

Reviewer's Responses to Questions

**Comments to the Author**

1. If the authors have adequately addressed your comments raised in a previous round of review and you feel that this manuscript is now acceptable for publication, you may indicate that here to bypass the “Comments to the Author” section, enter your conflict of interest statement in the “Confidential to Editor” section, and submit your "Accept" recommendation.

Reviewer #1: All comments have been addressed

2. Is the manuscript technically sound, and do the data support the conclusions?

Reviewer #1: Yes

3. Has the statistical analysis been performed appropriately and rigorously? 

Reviewer #1: Yes

4. Have the authors made all data underlying the findings in their manuscript fully available?

Reviewer #1: Yes

5. Is the manuscript presented in an intelligible fashion and written in standard English?

Reviewer #1: Yes

6. Review Comments to the Author

Reviewer #1: (No Response)

7. PLOS authors have the option to publish the peer review history of their article (what does this mean?). If published, this will include your full peer review and any attached files.

Reviewer #1: No

---

## [Editor Report · Acceptance letter]

2 Dec 2021

PONE-D-21-23534R1 

3,3'-Diindolylmethane induces apoptosis and autophagy in fission yeast 

Dear Dr. Ueno:

I'm pleased to inform you that your manuscript has been deemed suitable for publication in PLOS ONE. Congratulations! Your manuscript is now with our production department. 

Kind regards, 

on behalf of

Dr. Reiko Sugiura 

Academic Editor

PLOS ONE